# Experiencing Climate Change and Living Through It—Provocations for Education Based on South African Youth Experiences of Climate Change Policymaking and Politics

Tyler Booth [1] and Harriet Thew [2,*]

1   School of Earth and Environment, University of Leeds, Leeds LS2 9JT, UK; tylerbooth38@gmail.com
2   Sustainability Research Institute, University of Leeds, Leeds LS2 9JT, UK
*   Correspondence: h.thew@leeds.ac.uk

**Abstract:** This research investigates youth participation in climate change politics and policymaking in South Africa, responding to a notable lack of Global South-facing studies in the literature on youth climate activism. Guided by our lead author's substantial engagement in South Africa's youth climate movement from 2014–2024 and drawing upon semi-structured interviews with 12 young climate activists, we offer rich insights into young South Africans' motivations to participate in climate politics and policymaking. We then draw upon these insights to offer a series of provocations for climate change education. On investigating why youth participate, we find that although they report similar intrinsic and extrinsic motivations for participation to their Global North counterparts, South African youth climate activists place far greater emphasis on situated awareness and lived experience. We further improve the understanding of how young people perceive meaningful participation and climate (in)justices and how this shapes and is shaped by their activism. We therefore emphasise the value of incorporating both local case studies and affective elements in climate change pedagogies to encourage participation in collective climate action. Ultimately, we call for an enhanced recognition and inclusion of youth as active contributors to, and educators within, climate change governance and for the reconceptualization of youth climate activism, and policy engagement as key sites of transformative learning.

**Keywords:** youth climate activism; climate change education; youth participation; motivations for activism; intersectionality; transformation-oriented learning; South Africa

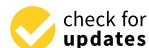

## 1. Introduction

The greatest challenges facing present and future generations are the interlinked planetary crises of climate change, biodiversity loss, and pollution (UNFCCC, 2022). As a multi-scale, multi-level and multi-actor problem, climate change governance is incredibly complex (Jänicke, 2017) with impacts and vulnerabilities extending over space and time. This raises important questions of equity and justice which, along with the limited technical and financial capacities of governments (state actors) warrants alternative governance mechanisms (Page, 2006; Hickman, 2016) including a greater inclusion of non-state and sub-state actors (Acuto, 2010; Andonova et al., 2009; Gordon & Johnson, 2017; Keck & Sikkink, 1999).

Young people are an important yet often overlooked sub-grouping of non-state actors (NSAs) in climate change governance. Despite often having limited physical resources, young people bring new capacities, innovative ways of thinking, and alternative approaches

to organising, which can enrich climate governance. Much of the literature on youth in climate change governance focuses on the role and experiences of young NSAs in the Global North and within the United Nations Framework Convention on Climate Change (UNFCCC), with limited attention to the Global South (Andersson, 2017; Beukes, 2021; Bowman, 2019; Elsen & Ord, 2021; Nkrumah, 2020). Their visibility within the UNFCCC and beyond remains limited despite a recent, substantial expansion in the global mobilisation of youth-led climate activism (Della Porta & Portos, 2023; de Moor et al., 2021; Han & Ahn, 2020; Islam, 2023; Yona et al., 2020).

Youth participation in climate change policymaking and politics is often motivated by perspectives of climate (in)justice (Kotzé & Knappe, 2023; Morgan et al., 2023) with youth claims of intra- and intergenerational justice identified by several scholars (Aiello & Di Martino, 2024; García-Antúnez et al., 2023; Ursin et al., 2021). Regardless, youth still receive less attention than the experiences of other NSAs, a gap which is even more pronounced at national and sub-national levels (Luckett, 2022; O'Brien et al., 2018; Thew, 2018; Thew et al., 2020; Vogel et al., 2022). Resultantly, there is limited understanding of what effective and meaningful youth participation in climate change governance looks like in climate policy, further exacerbating the potential for intergenerational injustice (Kosciulek, 2020).

South Africa presents an interesting case: the South African Constitution and Bill of Rights explicitly state that all citizens have the right "to an environment that is not harmful to their health or well-being" and "to have the environment protected, for the benefit of present and future generations" (The South African Constitution, 1996, p. 9). Despite this, the country remains the second most unequal country in the world, the largest carbon emitter in Africa, and the 14th largest globally (Carbon Brief, 2018; Chersich & Wright, 2019). Climate change is set to deepen inequalities and vulnerabilities running across socioeconomic, racial, and gender lines. The latest Intergovernmental Panel on Climate Change (IPCC) report indicates that climate impacts for Southern Africa include periods of increased and decreased rainfall leading to floods and droughts, an increased risk of tropical cyclones, and consequential food and health insecurity (Trisos et al., 2022). Tackling pre-existing inequalities is thus a necessary precursor for preventing further climate-related injustice (Chersich et al., 2018; Tanner et al., 2022).

### 1.1. Research Questions and Main Contributions

This research investigates youth climate activism in South Africa, exploring the participation of young people in national and sub-national climate policy and political processes to address the following research question:

What motivates youth climate activism at the national and sub-national levels in South Africa?

We contribute to the research on youth climate activism in several ways:

(1) We expand the empirical literature to address the geographical gap pertaining to African experiences specifically and Global South experiences more broadly. (2) We address the methodological gap by contributing rich qualitative data which interrogates youth-led participation spanning a range of different youth organisations and opportunities rather than focusing on one network or policy forum. (3) We apply, analyse, and visually depict a series of intrinsic and extrinsic motivations to increase the understanding of why young people participate in climate policy and politics. (4) We improve the understanding of how young people in the South African context perceive climate related (in)justices and how this shapes and is shaped by their activism. (5) We expand on existing work on the links between participation in policymaking and activism and transformative learning and its implications for climate change education. Underpinning each of these contributions is our

emphasis on young people's agency and their relevance as central stakeholders in climate governance who are already experiencing and responding to a climate-changed world.

### 1.2. Defining Youth in the South African Context

This study follows the South African National Youth Policy (NYP) in defining youth as 14 to 34 years (Department of Women, Youth, and Persons with Disabilities, 2020). Persons aged between 0 and 34 (children and youth) make up 64% of the population which means that, in addition to the burden they face as the generation who's expected life-spans overlap with medium to longer-term timescales of dangerous climate projections, they constitute the demographic most affected by climate change in South Africa (Beukes, 2021; Statistics South Africa, 2022). Notwithstanding the current social and environmental burdens facing young people, youth unemployment rates are much higher than the national average, with the 15–24 age range experiencing a 63.9% unemployment rate (Statistics South Africa, 2022). As the living generation that will be most affected by climate change with the least political power (Yona et al., 2020), young people deserve multiple seats at the decision-making table. In this study, we focus specifically on young people who strive to represent "youth" as an interest group in climate change policymaking and activism, i.e., emphasising the experiences, needs, and rights of their peers and of future generations, rather than those who may be young but participate as representatives of their government, employer, or another civil society interest group.

### 1.3. Understanding Youth Participation, Climate Change Governance, and Climate Justice

The literature on youth participation in climate change governance can be grouped into four distinct yet overlapping research agendas: studies of youth policy participation in the UNFCCC through the constituency of Youth NGOs (YOUNGO) (Kolleck & Schuster, 2022; Thew, 2018; Thew et al., 2020, 2021; Yona et al., 2020), studies of youth political participation in climate actions or protests outside of institutional decision-making settings (Bowman, 2019; de Moor et al., 2021; Nissen et al., 2020; Wahlström et al., 2019) and studies relating to climate and energy literacy and the role of pedagogy in effective youth engagement and mobilisation (Cutter-Mackenzie & Rousell, 2018; Gladwin et al., 2022; Karsgaard & Davidson, 2023; Komatsu et al., 2023; Kowasch et al., 2021; Rappleye et al., 2024; Svarstad, 2021). The fourth emerging body of literature studies climate litigation as a means for which young people can contribute to climate governance, often through a lens of intergenerational justice (Donger, 2022; Dozsa, 2023; Nkrumah, 2021b; Parker et al., 2022; Steinkamp, 2023). This present study sits within the youth participation literature related to education and the role of pedagogy in effective youth engagement and mobilisation.

This literature is limited by its reliance on quantitative methods which are ill equipped to capture the rich, complex, and heterogeneous experiences of youth climate participation. It is further limited by its focus on one youth network (i.e., FFF) and its limited geographic perspective which rarely expands beyond cities in the Global North (Neas et al., 2022). However, it highlights some useful conceptual tools, such as Kowasch et al.'s (2021) application of the psychological conceptualisation of intrinsic and extrinsic motivations to youth climate activism, which this paper also utilises (Kowasch et al., 2021).

Clear links between participation, justice, and vulnerability are evident in the study of climate change governance. As the effects of climate change are distributed disproportionately and more strongly felt by those with pre-existing vulnerabilities, particularly in the Global South, proactive efforts to prioritise justice in climate change governance is required. This includes attention to historic responsibility and injustices as well as to the air distribution of benefits in the present and future (Dryzek et al., 2013; Gardiner, 2013). Another dimension to this argument is that actions of individuals or groups of people

in one part of the world affect people beyond national and temporal borders. Therefore, duties, obligations, and responsibilities constitute the concerns of international justice (Baer, 2011). Similarly, intergenerational justice links to the timescale of effects, and the fact that todays' young generation and the future generations they will birth will experience the brunt of climate impacts (Ursin et al., 2021; Howarth, 2013).

Justice within the youth participation literature is considered from two angles: whether/how their participation results in procedural and representation justice, and the nature of youth justice claims (Fraser, 2014; Han & Ahn, 2020; Huttunen & Albrecht, 2021; Thew et al., 2020; Schlosberg, 2004; Scott & Malivel, 2021; Ursin et al., 2021). Within the International Youth Climate Movement (IYCM), youth protestors and activists have called for 'climate justice now' and 'systems change not climate change'. Their claims include both inter and intragenerational justice, as well as social justice as integral to climate justice (Conner et al., 2023; Cugnata et al., 2024; Han & Ahn, 2020; Pavenstädt, 2024; Ursin et al., 2021). Here, we investigate youth perceptions of (in)justice.

Mainstream ESD approaches have been critiqued for framing climate change as a purely scientific problem which limits proposed solutions to the technocratic (e.g., R. Khan, 2008; Muccione et al., 2025) and centres white, middle-class priorities such as biodiversity conservation above addressing socio-environmental injustices which disproportionately affect the working classes and people of colour (F. Khan, 2000).

Climate change activism as a form of experiential learning is argued to be a way to expose learners to alternative discourses which critique these technocratic and colonial framings while providing opportunities to challenge the status quo (Kowasch et al., 2021). Several scholars have suggested that climate change activism should therefore be considered as an integral part of Education for Sustainable Development (ESD) which can build the capacity and confidence of young people to act as environmental citizens and contribute to transformative socio-ecological change and to advance climate justice (Kowasch et al., 2021; McGregor & Christie, 2021; Neas et al., 2022; Trott, 2024). There is some hesitation around this due to perceptions of violence associated with activism; however, a recent South African study found that engagement with climate activists as a part of an educational programme was celebrated as providing an integrated, embodied experience which engaged learners' emotions, creativity, and spirituality, providing a joyful and healing safe haven from "violent" national politics.

Another recent study in the USA found that young climate activists play dual roles as learners and educators: learning how social movements operate while educating others on issues of climate justice (Trott, 2024). This warrants a closer investigation across different contexts as it could offer significant potential for transformative change in challenging intergenerational power dynamics within education.

Reviewing the post-2018 literature on youth climate activism, Marquardt et al. (2024) suggest that although rapid quantitative studies offer an entry-point to this field of study, a step-change is required with research that centres young people, challenges adultism, and goes beyond the misleading conceptualisation of youth as a homogeneous group (Neas et al., 2022). Studying youth-led climate activism in Global South contexts is particularly important to move the field beyond theories generated through a narrow focus on mainly white, economically privileged participants. This is necessary to capture the diverse perspectives, experiences, and contributions made by young climate activists around the world, particularly considering calls to question western universality and counteract the impacts of coloniality in the climate movement (Karsgaard & Davidson, 2023; Ritchie, 2021). Furthermore, recent research suggests that young people's perceptions of climate change and willingness to act vary between countries. A review of 51 studies covering 20 countries (Lee et al., 2020) found that young people in Australia, the USA, and the UK

were less concerned and less willing to act on climate change than their counterparts in other countries. The disproportionate number of studies in their review which focus on nations ranked as "High Income" by the World Bank (14 out of 20) and the absence of any African studies is a further cause for concern.

Very few works have analysed national or subnational youth climate policy participation, though exceptions include Nilan (2021), Fernandez and Shaw (2013), and Wilf et al. (2024) in Asia, Castañeda (2023) in Latin America and the Caribbean and Vogel et al. (2022), Benkenstein et al. (2020), Zimba et al. (2021), Luckett (2022) in Africa. As a result, investigating youth activism in the Global South could offer rich insights for educators seeking to enhance and support young peoples' capacities and motivations to meaningfully engage with climate policy. Based on our understanding of previous studies on youth climate activism, we are particularly interested in mechanisms that enable collective action and participation and challenge neo-liberal pedagogies that promote individual behaviour change which is often more incremental than transformative (Lotz-Sisitka et al., 2015; McGregor et al., 2018; Trott, 2021; Verlie & Flynn, 2022).

In the South African context, a handful of studies have investigated youth climate political and policy participation. Beukes (2021) highlights the important role of youth as game-changers for climate action, offering an overview of some youth-coordinated activities. However, the study is arguably methodologically limited, based on a brief document analysis with recommendations made despite a lack of qualitative engagement with young people. Nkrumah (2020, 2021b) explores youth political participation through legal activism, recommending climate litigation. Whilst his study aims to offer insights of "how [youth] could mobilise themselves into a Youth Climate Movement", (Nkrumah, 2020, p. 358) it does not ground these in any existing networks or coordinated activities that youth engage in. Beukes and Nkrumah both suggest that youth experiences of climate activism are varied, levelling critiques of exclusion and tokenism which demand further study (Vogel et al., 2022; Nkrumah, 2020). Vogel et al. (2022) make a first step towards this, applying a youth participation framework which offers insights into a youth-led policy engagement process at the sub-national level. They suggest that youth policy participation cannot be pigeon-holed and contribute critical evidence showing that how a process is initiated and facilitated has significant bearing on its sustainability and effectiveness. Further research should explore this, looking beyond the single process they studied, to avoid the mistakes made in the earlier literature of homogenising youth participation. Finally, Luckett (2022) focuses on youth activism in the urban periphery of Lephalale, a community born around the coal industry. They untangle and analyse the complexities of youth activism and the relationship between youth (un)employment and environmental (in)justices where young people engage in invited and invented spaces at the political level, rather than a policy level. This demonstrates the need for research into young people's perceptions of climate-related (in)justices in South Africa and how these shape and are shaped by their complex participatory experiences.

## 2. Materials and Methods

### 2.1. Research Strategy

Qualitative research aims to explain, understand, and elucidate the meaning of phenomena through words and graphics as opposed to numbers and statistics (Bryman, 2016). Correspondingly, a zigzag method based on the initial inductive coding of data, followed by further refinement and abductive coding based on relevant literature was used to develop our research questions (Emmel, 2013). Zigzagging is a methodology that emerged from the critical realist paradigm which involves crafting a reflexive conversation between data and theory (Emmel, 2013). It sets out to identify causal mechanisms (guided by and to

develop theories) from empirical evidence, drawing on practices from realist and feminist philosophies. This type of research is particularly relevant to the study of justice and power-ladened participatory experiences in its intentionality to be non-hierarchical, to acknowledge multiple valid perspectives, and to be attentive to researcher positionality and the ways in which an insider perspective can shape research findings.

### 2.2. Positionality

At the time of data collection and analysis, the lead author was a member of the youth climate activist community in South Africa. This meant that she was positioned as an 'insider' of the group she was investigating (Dwyer & Buckle, 2009) and had pre-existing working and/or friendly relationships with many of the participants. Acknowledging that this could shape what the participants did and did not disclose (Bryman, 2016) the lead researcher regularly reflected on her interactions (Dodgson, 2019) and was cautious to ensure that participants did not assume familiarity, encouraging them to explain their perspectives and experiences fully (Breen, 2007; Dwyer & Buckle, 2009) while striving to maintain an approach which was reflexive, non-hierarchical, and non-manipulative (Cotterill, 1992).

As a cis-gendered, white, young, South African female from a middle-class background conducting research with a group that is largely led by people of colour from different cultural, ethnic, and socio-economic backgrounds, within a country that has a history of racism, it was necessary for the lead author to remain attentive to the power dynamics of race and racism that remains prevalent in South Africa (Variava, 2020). To mitigate this, she remained reflexive of her positionality, did not claim to know participant's stories, utilised a self-disclosive interview strategy that tackles power asymmetries (Abell et al., 2006), and made clear what participants' data would and would not be used for. The second author is a cis-gendered, middle-class white female academic based in the United Kingdom. She was not involved in data collection and was careful not to ascribe meaning to data, ensuring that all interpretation was made by the lead author. As far as possible, results are presented in the participants' own words so that any interpretation from the research team is transparent and open to reinterpretation by the reader.

### 2.3. Sampling and Data Collection

Participants were selected using purposive sampling based on "the researcher's experience or knowledge of groups to be sampled" (Lunenburg & Irby, 2008, p. 175) as well as snowball sampling based on recommendations from recruited participants. While there was no set checklist for inclusion criteria, youth that had taken on leadership roles (formally and informally within youth organisations) and who had led or co-designed a policy or political process, or who at the time were actively engaged in the South African Just Transition Framework process, were approached for interview. Additionally, the lead author approached young people across different climate organisations, gender identities, and localities represented in the South African youth climate movement. This sampling method is designed for research credibility rather than for generalisability (Bryman, 2016; Lunenburg & Irby, 2008) which we remind readers to be mindful of when considering youth activism in other Global South contexts. Twelve participants took part including five female, five male, and two non-binary young people from five South African provinces.

Data collection consisted of semi-structured interviews. Self-disclosive, semi-structured, open-ended interviews (Mason, 2017) introduced a conversational approach allowing the interviewer to react to information shared while striving to reduce power asymmetries (Bryman, 2016; Kvale, 1996). All the interviews were conducted between June and August 2022, at a time suggested by the participant. As the lead author was

based in the United Kingdom at the time of data collection, all interviews were conducted online over Microsoft Teams (11 participants) or Zoom (1 participant). Typically, the interviews lasted between 45 min and 1 h. All participants consented to their interview being recorded and transcribed verbatim. Due to loadshedding and uneven network ranges in South Africa, around half of the interviews were conducted without video to spare data and bandwidth. While de Villiers et al. (2021) argue that this limits access to non-verbal communication, as an insider researcher who is familiar with the participants and context, this was negated through the detection of verbal cues and additional probing to clarify the responses from the questions asked. Pseudonyms have been assigned to participants that are culturally and gender appropriate to protect participants identity. See Table A1 for demographic data of participants—while participants were all active participants of one or more youth organisation, their affiliation has not been disclosed to protect their anonymity. See Appendix B for the interview guide used in collecting the interview data.

### 2.4. Data Analysis

For the interview data, Nvivo 14 software release 1.6.2 was used to conduct abductive coding. The coding framework was guided by youth participation frameworks examined in Section 1 and emerging ideas from interview transcripts in line with the zig-zag approach (Emmel, 2013). Abductive coding of interview data was guided by themes from the literature, including "barriers", "climate education", "climate justice", "meaningful participation", and "narrative". The analysis of "climate justice" began with themes identified in the literature—intergenerational, participation, procedural, recognition, and representation justice. Following an inductive approach, the lead author further explored how youth defined climate justice, expanding the codes to include additional themes such as energy, environmental, gender, racial, social justice, and systems change. Finally, "narrative" codes were subcategorised into "intrinsic" or "extrinsic" motivations, using a zig-zag approach between the data and existing literature.

### 2.5. Limitations

This study is limited temporally as the collection of data for this research was conducted between June and September 2022. Due to institutional ethical restrictions and time limitations, it was not possible to include youth under 18 in our study despite our intention to align with the definition of youth as outlined in the South African National Youth Policy. Despite this national definition, we found that several youth organisations engaging in climate policy and activism have an upper age limit of 30, rendering it challenging to recruit older participants. Our study is also limited geographically in its focus on South Africa, though this was necessary to expand the focus beyond the global level to include national and sub-national participation and we hope that our theoretical findings may be applicable to other contexts. Our sample size is small, and while although not uncommon in qualitative research of this nature, this does limit the generalisability of our results.

### 2.6. Ethical Considerations

Ethical approval for this project was given by the University of Leeds (reference AREA 20-070). A process of informed consent was followed which provided prospective participants with an information sheet detailing the aims, objectives, methods, and data handling procedure which they agreed to in writing. Confidentiality was assured throughout the lifespan of the project, though due to the purposive sampling methods, for some individuals that were involved in or led projects, only a handful of young people had specific high-profile experiences, meaning that participants were informed that their anonymity could not be guaranteed, as it was possible that they could be identified by their peers.

## 3. Results

Our results shed light on young people's reasons for, and experiences of, climate activism in the South African context. In Section 3.1, we analyse why young people participate, exploring their intrinsic and extrinsic motivations, and in Section 3.2, we consider the barriers to meaningful youth participation.

*3.1. What Motivates Youth Climate Activism at the National and Sub-National Levels in South Africa?*

Routes into youth climate activism are diverse. One-third of the youth interviewed cited lived experiences of climate change and other environmental crises. During their formative years, they had encountered climate impacts such as floods or droughts or grew up near sources of environmental degradation including mine dumps or poorly managed waste disposal areas. Around half of the participants found routes to engagement in climate action through involvement in school clubs or NGO-led education initiatives. Several others entered the climate movement from other forms of social activism, such as gender, health, and democracy building.

Our participants articulated a wide range of motivations for their climate activism, including intrinsic motivations, i.e., engaging in something because it feels good or satisfying and extrinsic motivations, i.e., engaging in something to achieve a desired external outcome (Ryan & Deci, 2000). These are explored in turn below along with the links between them, before being depicted in Figure 1.

### 3.1.1. Intrinsic Motivations

Previous studies of youth climate activism have cited a general concern for the environment and an awareness of climate change as an intrinsic motivation. Our participants cast a different light on this, with almost all participants emphasising the importance of *context-specific* awareness, often based on lived experience:

> *"When I got to varsity, they had what they called the Youth Policy Committee (YPC) . . .They were divided in different thematic areas, and I was part of the gender one at first. I remember very well. . .there were not enough people in the climate change working group, even though it was the first one. It wasn't really interesting to me. . .I didn't know what that was, to be quite honest. Then I didn't know how I fit in and all of that. . .'cause for the longest time, I thought anyone speaking on climate change is probably vegan, probably stayed with monkeys in a forest and was studying them and stuff. And I was like, yeah, I know, I don't think I'm interested in any of that. But then I just remember, you know, watching a video on YouTube. . . that was speaking about our houses burning and. . .what climate change looks like in South Africa and what that means for a lot of South Africans. And I think after listening to that, I got intrigued".*—Nolwazi

Participants reported witnessing generalised environmental concern from other young people in South Africa but emphasised that her motivations were underpinned by context-specific awareness and connecting to the realities of their peers and communities:

> *"We saw the, you know, the fee-paying schools, which are usually working middle class. . .they would come and dominate the discussion and be like 'we should protect the oceans (giggles), we should protect. . .the aquamarines, we should protect life under water.' And I'm not saying we shouldn't be protecting that. But then there was also an issue of a young person who was in rural Eastern Cape was struggling to just get water".*—Nolwazi

> *". . .climate education that we were working on [is] very accessible to the realities on the ground. And we didn't much want to, kind of, talk about the polar bears, because what do polar bears have anything to do with the South African context?"*—Daniel

Kgosi explained how he became involved in climate activism after engaging with a project which aimed to support a disadvantaged community of waste reclaimers living in a waste disposal area:

*"In 2014…when in the community that was living in there, there was a lot of deaths, so I saw this as an issue and as a young kid I did not know really what to do, where to start, other than to start, like, a high school club where we speak to the Principal about the issue…and then by chance, the same year, Youth@SAIIA were hosting a [project]… and since then I've been part of other youth groups and [organisations]".*

This strong focus on lived experiences and resultant awareness of intersecting vulnerabilities and injustices experienced by local communities was another common motivation, with several participants citing exposure to/participation in other social movements as a precursor to their climate activism:

*"I come from an activist family… from an understanding [of] societal issues".*—Anaya

*"My climate activism is relatively new. I started in 2018. It started with, there were xenophobic attacks in the country where basically foreign nationals from other African countries were being attacked by South African locals. And so I attended a public lecture…out of interest, for social issues and got a chance to, like, engage with different people from academia, different stakeholders and government…People from the African Union which looking back now was really a significant part of my activism journey".*—Themba

*"I started as a social activist, focusing on issues of HIV prevention… then I also became an activist for issues of international humanitarian law… then shifted the focus and started in the environmental justice or climate justice movement".*—Mthunzi

Many participants were motivated by the perception that climate action was insufficient and either highlighted this explicitly or connected it to their affective response to lack of action, again revealing that context-specific awareness plays an important role:

*"Well, my real activism started probably when I got aware about the climate crisis, so that was 2019. I was attending the model UN that SAIIA hosted, and I think if memory serves correctly, there was a video that was played…which was explaining what the climate crisis would look like in sub- Saharan Africa and specifically in the South African context. What it looks like now and what is going to happen in the future. And I remember being taken aback by what I was seeing that there was not really a lot being done about climate change".*—Tumelo

Young people's context-specific awareness of climate change, including their perceptions of inaction and injustice, affective/emotional responses such as frustration, anger, and overwhelm, which half of the participants noted as a key motivation both for themselves and for others:

*"I attended one of the global strikes and it was in Joburg, and I wanted to take part in activism… and then in 2020, I basically lost my temper with the world and said that I'm going to boycott school on Fridays".*—Rania.

*"I think by and large, young people are just frustrated and angry that the system has not been working for them. When you have a 65 plus unemployment rate in the youth category, I think we are speaking of a society that is in perpetual crisis, perhaps because this has been consistently over many years…we need to be able to understand the relations of power and how it affects communities and people on the ground…[we need] the kind of forces, or powershift, within the ruling party and within other political parties that showcase that the issue of climate is on the agenda, but it's one that's being, not being*

*collectively agreed upon. . .we've seen there's a huge desire for this. We've seen huge interest in young people wanting to work together to solve some issues".*—Amir

*"I think lived experience is something that's really important and it's and the way that you interact—action is not through policy, it's not through a policy briefing. It's through lived experience, right? So. . .the way that you're going to get someone to get up and act is, you're gonna say the fact that bread is expensive, or whatever it is the thing [that] isn't growing well. . .. you know, there's a drought, so nothing's growing. If you explain that this is going to continue to happen, that's when someone that's emotive, you know, someone is going to get angry about that and [they are] going to act".*—Anaya

*"It always feels like you better off to address climate change when you are equipped, rather than when you are not"*—Kgosi.

Access to participatory opportunities, including through formal and non-formal education, was also mentioned by several participants as an important motivation for youth climate activism. These intrinsic motivations therefore appear to be mutually reinforcing, with participatory opportunities creating spaces for young people to reflect on their lived experiences, context-specific awareness and affective responses while offering routes to further learning and action.

*"My school was privileged enough to participate in model UN, and then we were surrounded or, we always thought about big key things happening around the world, so got exposed to [the] issue of climate change one, but also experiencing effects of climate change, and seeing by family in Limpopo experiencing the effects of climate change. Living through them and trying to also adapt and find ways to live through it. And so that's primarily what drove me, that was my intro to the space".*—Lerato

*"I was immediately overwhelmed but then I. . .saw the opportunity to join the 2019 YPC that [South African Institute of International Affairs] (SAIIA) has and so that's where I was thrown into what climate change was, what the actual climate crisis was, and how we respond to it".*—Tumelo

*"The actual devastation and damage occurs across the continent firstly and then across the world, it became very clear, especially around the times that we saw massive wildfires, especially around the time we started hearing about this idea of day zero. . .Even back then when I was in the final years of high school, we had those conversations about, what if we had a drought? What if we had this? What if we had that? And so, when I started being a part of the environmental club, I learned a lot more about those things".*—Daniel

Some participants were also motivated by self-confidence in their skills and agency. Here, the relationship between intrinsic and extrinsic motivations was particularly apparent, with clear links between their positive experiences of growing self-confidence and the collective formulation of constructive hope, bolstering young activists' belief that their actions (both individual and collective) could make tangible differences:

*"I don't do science. I couldn't help on the science side of things and with terms of policy like I it, I find it really difficult to read those documents and attend these meetings and not completely, like zone out and shut down. But what I figured that what I can do and what I know how to do best is art. And so, I wanted to try and help and do activism using art as a medium. And so, I began to plan a performance art protest to raise awareness about the climate crisis. And so that's where my activism started".*—Rania

*"This was a moment of hope where. . .thousands (or if not hundreds) [were] of young people out in the streets of Pretoria demanding justice and change".*—Amir

The positive experiences of unity and community encountered in participatory spaces also acted as intrinsic motivation, while fostering a collective sense that their participation

was a beneficial outcome, given the need for increased youth representation in socio-economic decision-making:

> *"SAIIA in their programmes opened me up to like a diverse group of young people from all across the country, from different backgrounds, different schools, different everything, and so that was really big learning curve for me. . . really allowed me to feel this was a sense of urgency in myself to say that it's really important that I am as a young person advocating for social issues that I see in my community and that as a young person, I take a stance and making sure that I involve myself in and becoming not only conscious about what is what is the socioeconomic context around me but also how what can I do, what role can I play to advocate or be a catalyst for change?"*—Themba

3.1.2. Extrinsic Motivations

Extrinsic motivations highlighted by our participants included the pursuit of political, social, and environmental outcomes, ranging from incremental political actions to radical socio-economic reforms. Rania demanded *"for the Department of Environmental Affairs to declare a climate emergency"*, meanwhile Lerato emphasised the following:

> *"I feel personally that climate policy, I'm engaging in climate policy, is very critical in order to shape. . .implementation on the ground. . .And I also needed a way to hold government accountable because we know government is there to ensure the well-being of people".*

While Daniel felt that, as follows: *"Youth have been the call for change time and time again and I think right now we see it even more clearly and that the only way forward is not to call for economic justice or social justice, or environment—is to call for an entire system reform. It's calling for the system to be uprooted".*

Many participants were motivated by a strong desire to challenge practices they perceived to be unsustainable and unjust. This included perceptions of intergenerational injustice:

> *"Young people will get to make decisions for our future because we are the ones we have to face the consequences and then have every person be involved in this space in whatever way they can".*—Kgosi

All youth participants consistently emphasised the importance of inclusive, intersectional approaches to climate justice, striving to address social injustices through the meaningful inclusion of groups marginalised by their gender, race, and indigeneity. An increased representation of young people in the climate movement was a key extrinsic motivation:

> *"We can't afford to have young people give up on this. This is a fight that they must be a part of".*—Nolwazi

However, many participants also demonstrated a strong understanding of intersectionality and a commitment to improving the inclusion of other groups.

> *"The climate space didn't have enough black voices, especially black female voices. . . in the climate space or the climate discourse, women were seen as vulnerable. . .and it was left at just that. . .[they] had to be protected by everyone, you know. Umm, but that was funny because the more I read up on climate change and Africa, I realized that women were truly at the forefront of, you know, climate mitigation and adaptation when it came to climate change, because they lived in these communities and they saw the little changes in the environment".*—Nolwazi

> *"Before we did this program and training many of [the communities my organisation works with] which are often working class, lower income, largely African thought the climate, the climate crisis to be a white issue. A white person's issue. So that idea, of race, then featured. . ..we had to break those notions. . .. when we do antiracism work,*

*we understand racial justice is interconnected to climate justice, social justice, economic justice and so on".—Amir*

*"[linking] Climate change to bread-and-butter issues and making sure that people understand the effects...Nobody cares that the temperature's increasing, but they care that because the temperature is increasing, you're not going to have water, or food, or your housing is not going to be withstand the flood or whatever it is".—Anaya*

Rania and Themba reflected on the need for intersectional representation:

*"My activism kind of changed its focus from being about spreading awareness about the climate crisis to being intersectional and inclusive, fighting for inclusivity and making sure that no one is left behind because in these political spaces, it's very often governments and ministers deciding what is fine. My activism centres around engaging in climate policies and making sure not only that young people are engaged, especially in climate policies, but also that youth voices are represented, especially those of women".—Rania*

*"When [young people] are included, we not only fight for our inclusion, but it becomes a fight for inclusion of other marginalised groups... when you empower young people and amplify the voices of young people, they also bring in the voices of other marginalised communities such as women and grassroots communities".—Themba*

Participants further emphasised the importance of inclusivity asking the following:

*"So how do we make it more accessible for indigenous voices? That is again the state because the state is the only mechanism that I've identified that can do it, going and actively bringing in people from indigenous communities to also be part of these discussions".—Tumelo*

*"But young people do want to engage. It's just giving them the space and opportunity; I think to do so and ensuring that the systems that do govern us are geared to respond to their needs and not push them to the wayside".—Amir*

Interestingly, almost all participants emphasised the concepts of a just transition and intersectionality as central to their demands, regardless of their varied motivations and routes into youth climate activism.

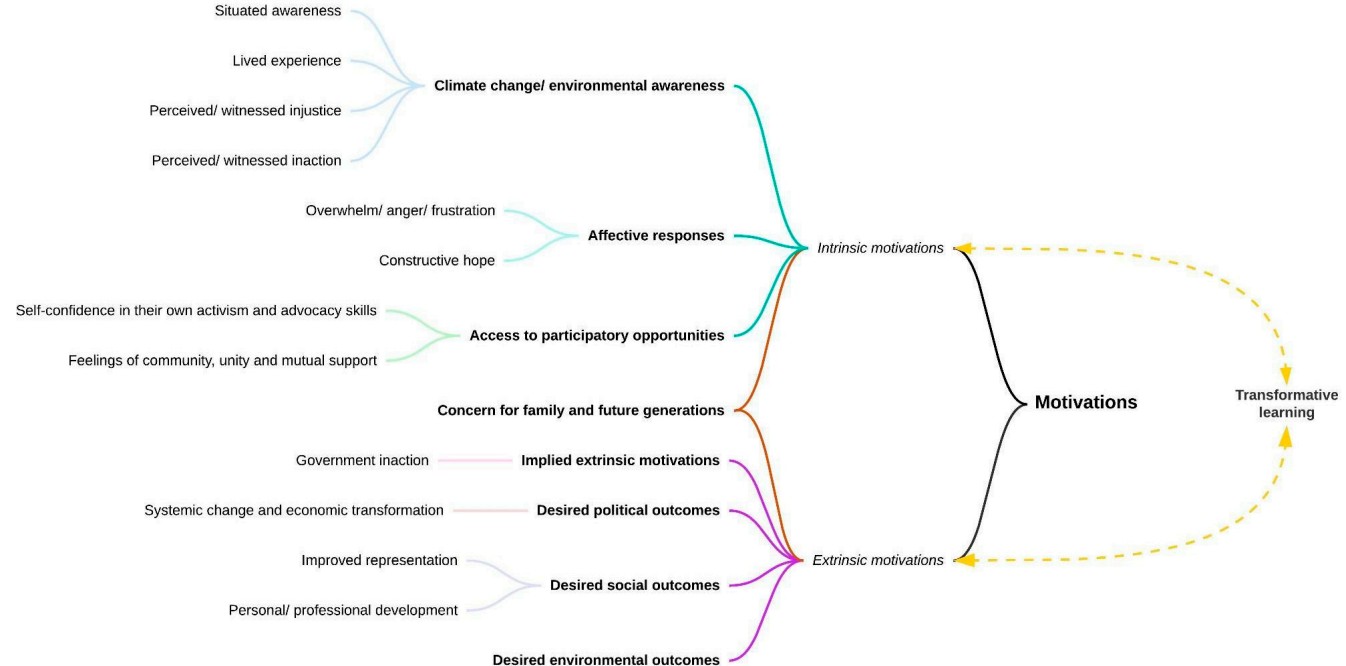

**Figure 1.** A representation of our findings on youth motivations for climate activism.

Section 3.1 has explored young people's intrinsic and extrinsic motivations for participation in climate activism in South Africa. These motivations and the links between them are synthesised and depicted in Figure 1. This Figure also strives to capture the relationship between motivations and transformative/transgressive learning which we understand to be changing perceptions of climate change and its solutions as well as changes to young people's identities (Lotz-Sisitka et al., 2015). This is based on our understanding that each of the motivations here could also be considered examples of transformative/transgressive learning. Developing climate change awareness based on context-specific understanding, lived experiences, perceptions of injustice, and perceptions of inaction are all shown in these results to change how these young activists understand climate change and themselves, and the affective responses of frustration, anger, and overwhelm all arguably make young activists receptive to transformative/transgressive learning and the formulation of constructive hope through access to participatory opportunities and the feelings of unity, community, and self-confidence again could all be seen as examples of young people undergoing a clear shift in how they perceive a climate changed world and their role within it.

A consistent narrative emerged of youth and their networks as key agents of change who demanded collective action and systemic change:

> *"We are tired and aren't going to be quiet!"*—Daniel.

> *"Young people are the ones standing up. . .[saying] this is the policy we need, or this is the alternative we need, or this is how we're going to do it".*—Anaya.

The participants also discussed key stakeholders, and their perceptions of vulnerability and blame based on unequal power dynamics and differentiated capacities. They favoured a justice framing when speaking of who should deliver climate action, with a vaguely referenced 'government' as responsible for the distribution of [climate] justice, while women, youth, and marginalised communities were seen as its primary recipients. Young people's perceptions of (in)justice were extremely important motivators for their participation. These are explored in Section 3.2.

### 3.2. Barriers to Meaningful Youth Participation

The inverse of what motivates youth, i.e., what demotivates youth has also been investigated as the barriers to meaningful youth participation—of which our participants identified many. Several participants spoke of tokenistic and limited engagement with youth. The youngest participant, Tumelo, explained that the first barrier was access, as a very limited number of people get access to the rooms where decisions are being made, with access predominantly granted though youth organisations. Lerato added that the way to gain access was "*to be on the database on the mailing list*". However, this can lead to a very narrow youth representation. Several participants raised concerns about youth appointments, where one young person is elected to a body to represent the voices of all youth. The most cited example was Ayakha Melithafa, who is the only youth commissioner on the Presidential Climate Commission.

> *"the pressure is on this one person to be talking on behalf of all these people. . .Ayakha has been one very good example of someone who's struggled with this. . .like, 'OK, well, now I'm supposed to speak for all these people. . .what mechanisms are there in place for me to represent [them]?'"*—Clara

A further limitation is the technical nature of policy language. This barrier is two-fold: firstly, climate policies involve extensive climate jargon, and secondly, the country has 11 official languages, but policies are only communicated in English. Rania, who refuses to

engage in policy for its lack of inclusion of marginalised voices, remarked *"why engage something you don't understand?"*

Ameena reflected on the need for education and capacity building to support youth participation:

> *"You know how much capacity young people are given within this space and how quickly they have to. . .build capacity and knowledge in order to engage and be taken seriously. . .. my experience with climate policies specifically has been the lack of educational structures and capacity building spaces that [support]. . .young people to actively engage with policy".*

Even when nominally provided, education and capacity building can be a barrier to meaningful participation, ladened with adultism and used to disregard young people's perspectives. After young people gain access into decision-making spaces and provide input or ask questions, Tumelo remarked that they are often met with a response of the following:

> *"Thank you for all that information, now let me actually educate you on what's happening".*

Similarly, Mthunzi shared a typical response when young people do raise their voice:

> *"Oh! the young person has spoken. Thank you very much. That was very powerful coming from a young person. . . They don't really care about what I have said, but what they care about is that I'm young and I've spoken. And to me, that's not being taken serious".*

Participants also highlighted the difficulties of navigating and belonging in the climate policy space. On one hand, overcoming imposter syndrome and feeling knowledgeable enough to enter and become active within youth groups and then engage with decision-makers was highlighted by Kgosi:

> *"Not everyone feels adequate enough to share their experiences".*

On the other hand, Tumelo explained the mental challenge of taking a break from activism, while still experiencing burnout: *"If I take a break, am I leaving a vacuum?"*. He added that "[you] *need a lot of optimism to be a climate activist".*

Another concern raised was that young people should not need to meet a threshold level of credentials to be taken seriously. Nolwazi mentioned the need to have the *"right credentials or the right MA from the right university"* and Kgosi exclaimed *"you don't need a degree to sit with Barbara Creecy [The former Minister of the Department of Forestry, Fisheries and the Environment (DFFE)]"*. Anaya instead referred to a generational divide suggesting that young people have a dissimilar struggle (of climate change) to the older generation who fought against Apartheid, they are perceived to lack the *"struggle credentials"* to be heard and respected as activists.

Even if an engagement was fruitful, and young people were able to share their inputs, several participants shared a similar expression to Themba who argued that, as follows:

> *"we have these meaningful conversations with stakeholders, but then we don't really get a follow up from there".*

Due to the lengthy turnaround of policies, youth participants had explained that it was difficult to gauge the extent to which their inputs were actively included, and whether they were included because youth raised them as opposed to civil society or NGOs raising them.

Clara elaborated on this perception of tokenised engagement, highlighting the need for adequate time to be dedicated to the participation and subsequent processes of feedback and accountability:

> *"I think often it's about having a token youth, which is like what you've said, where there's one young person who manages to, like, get into that space and then needs to advocate to bring more young people into that space. . . It also brings into question our*

*entire system. . .. it's [not enough to] . . .do a workshop and once the decision has kind of already been made. . .[say] we're gonna ask for people to respond and give feedback. . .the deadline is tomorrow. Then . . .feedback isn't incorporated in any inclusive way because they haven't allowed the time and the spaces for that".*

## 4. Discussion

*4.1. What Motivates Youth Climate Activism at the National and Sub-National Levels in South Africa?*

4.1.1. Intrinsic Motivations

Intrinsic motivations cited by our participants can be categorised broadly as an awareness of climate change and related environmental challenges, and affective responses to these challenges and the availability of participatory opportunities (as depicted in Figure 1). Although our findings share many similarities with the previous literature, showing that young people in South Africa share many motivations with their counterparts in other countries, there were some notable differences.

An awareness of climate change and other environmental challenges has been found to motivate young climate activists in the Global North with studies in Austria, Switzerland, Portugal, and Norway referring to a general expression of ecological concern (Haugestad et al., 2021; Kowasch et al., 2021; Martiskainen et al., 2020; Monteiro & Capelari, 2023). Our results suggest that although generalised environmental concern is felt by some young people in the Global South, the motivations for those who engage in climate activism reflect a more situated, context-specific awareness of climate impacts and other environmental hazards, based on lived experience. This difference may be because South Africa is already experiencing the effects of climate change, with multiple exposures to droughts, floods, wildfires, and heatwaves (Trisos et al., 2022; Chersich et al., 2018). This supports the argument that young people's concern about climate change and willingness to act is dependent upon whether one perceives that they are/will be personally affected (Lee et al., 2020).

It is not our intention to suggest that young people in the Global North are not already experiencing the impacts of climate change and related injustices, only that these experiences are yet to be sufficiently captured yet in the literature on youth climate activism (notable exceptions being studies of Northern indigenous youth) (Grosse & Mark, 2020; MacKay et al., 2020). This gap is symptomatic of the unequal distribution of environmental harms to underprivileged individuals and communities (Bullard et al., 2016) coupled with the methodological choices made in previous studies on youth climate participation and activism. The former, in its focus on international processes such as the UN climate negotiations where participation is limited to young people who have sufficient privilege to overcome the financial and other barriers to accessing the space (Marquardt et al., 2024; Thew, 2018). The latter, in its focus on major Global North cities where many participants have above average levels of privilege (Neas et al., 2022).

Similarly to previous studies (e.g., Brügger et al., 2020; Cologna et al., 2021; Cloughton, 2021; de Moor et al., 2021; Eide & Kunelius, 2021; Feldman, 2021; Kowasch et al., 2021; Ojala, 2012; Prendergast et al., 2021), we found that young South African climate activists cite affective intrinsic motivations. However, whereas young activists in Europe and Australia spoke primarily of fear and worry (Brügger et al., 2020; Cologna et al., 2021; Feldman, 2021) our participants spoke of frustration, anger, and overwhelm. Rather than being debilitating, these "negative" emotions were important factors in motivating and sustaining their activism. Rather than making generalised, anticipatory expressions of eco-anxiety, our participants closely connected these emotional responses to their situated awareness and lived experiences, further emphasising the importance of context-specific understandings of climate change. This demonstrates that different intrinsic motivations

interact rather than being stand alone and builds upon the literature on eco-anxiety and we reiterate the call made by Coffey et al. (2021) in their systematic review for future work to more closely investigate the link between eco-anxiety and lived experience.

In contrast with other studies, our participants did not mention guilt (Haugestad et al., 2021; Kleres & Wettergren, 2017), nor the anticipation of future guilt (Thew et al., 2022) as a motivation. This may be attributable to our participants' understanding of climate justice and responsibility at the international level, though future research of a qualitative, comparative nature is required to interrogate these differences.

Negative emotions were not the only intrinsic motivations identified. Hope was also very prevalent. Our findings emphasise the importance of constructive hope, rather than hope based on denial as a motivator for youth climate activism (Ojala, 2012). We build upon this in proposing that, when one's understanding of climate change is based on situated awareness and lived experience, hope based on denial is particularly difficult to sustain.

Constructive hope as a motivation for climate activism was often cited in tandem with references to young people's access to participatory opportunities. In addition to climate strikes, our participants highlighted the availability of policy engagement activities, conferences, local community projects, and formal and non-formal education projects facilitated by schools, universities, and NGOs in which young people were supported to research, communicate, and act on climate change. This supports Prendergast et al. (2021) in identifying "structural availability" as a motivation for youth climate activism (Prendergast et al., 2021).

Our results support the previous literature in confirming that youth climate activists are motivated when they feel self-confident about their own activism and advocacy skills as well as that of the groups they belong to (Kowasch et al., 2021; Neas et al., 2022; O'Brien et al., 2018). However, this debate is still live, with recent studies suggesting that neither individual/self-efficacy nor collective/group are statistically significant motivators of collective youth climate action (Haugestad et al., 2021; Prendergast et al., 2021). Haugestad et al. (2021) seek to explain this finding by proposing that young school strikers in Norway had not yet had an opportunity to experience being "impactful political agents". This is supported by their participants' assertion that participation in school climate strikes was the only way for young people to contribute to climate politics and policy because of their low status and lack of recognition in society. In contrast, our results show that despite young climate activists in South Africa also feeling that their generation lacks social status, they still report individual and self-efficacy as motivations for their initial and sustained participation. This could be because our sample included more seasoned climate activists or could be attributable to the structural availability of a broader range of participatory opportunities which provide opportunities for young people to have positive experiences at a younger age. That said, our participants also made several references to imposter syndrome, demonstrating the need for closer attention to the complex psychological factors shaping youth participation.

### 4.1.2. Extrinsic Motivations

We also identified a range of extrinsic motivations for young climate activism in this context. Our findings revealed various political motivations, including the desire for the Department for Environmental Affairs to declare a climate emergency, the need to hold the government accountable for climate policy, and calls for comprehensive system reform. These motivations align with the previous literature, which has documented a strong desire to challenge the political and economic status quo and to challenge unsustainable practices (Haugestad et al., 2021; Martiskainen et al., 2020; Monteiro & Capelari, 2023).

We also observed that participants were motivated by perceived confidence that they could contribute to a different future (de Moor et al., 2021; Kowasch et al., 2021; Monteiro & Capelari, 2023). This relates closely to the aforementioned intrinsic motivations of self-efficacy and collective efficacy which several studies have identified.

A recurring theme in our findings was the perception of government inaction and low trust in governments, as highlighted by studies from Cologna et al. (2021) in Switzerland, Feldman (2021) in Australia, Han and Ahn (2020) globally, and Pickard et al. (2020) in the UK. Whereas Bowman (2019), Cologna et al. (2021); Han and Ahn (2020), and Elsen and Ord (2021) found that youth climate activists were particularly motivated by the lack of institutional responses to climate projections and sought to encourage governments to "listen to the science", participants in South Africa emphasised that their motivation stemmed from the lack of institutional responses to climate impacts and their perception that governments hold responsibility for addressing injustice. This linked closely to our finding that young climate activists in South Africa recognise and are motivated by their recognition of the interconnections between climate change and other social and environmental movements, building upon previous work which has identified similar motivations in FFF strikers and participants in the UNFCCC (Kowasch et al., 2021; Monteiro & Capelari, 2023; Thew et al., 2022).

Climate justice was also a key theme that motivated South African youth, who expressed both 1st order claims of justice for themselves and as proxies for future generations, and solidarity claims. They predominantly framed climate justice in terms of advancing social justice, recognising intersectionality, and prioritising systemic change. Recognising the depths of historical inequalities, both in affecting their lived experience and those around them, they are taking concerted steps to prevent climate change exacerbating these. Thus, while Thew et al. (2020) found that youth claims had changed over time from 1st order to solidarity claims, this was in the context of youth from the Global North, whose primary motivation for their activism was unrelated to their lived experience as has been found here (de Moor et al., 2021; Han & Ahn, 2020; Thew et al., 2020). Despite the differences in methodologies, this remains relevant as it is unlikely youth have been co-opted (Nissen et al., 2020) to express either a first order or solidarity claim to justice, but on principle of their movement (intersectionality) to ensure all people are included from the outset. This concern for family and future generations thus presents itself as both an intrinsic and extrinsic motivation as depicted in Figure 1.

South African youth also expressed a similar intergenerational justice positioning as presented in Ursin et al. (2021). While South African youth expressed the four reasons why intergenerational justice is important—the current vulnerability to climate change, how they bear the brunt of future changes, the youth are the closest connection to the future, and their views are traditionally excluded—they further linked this to a generational divide of defining generational struggles (Ursin et al., 2021). Contextually, the current generation in power in South Africa lived through and fought against Apartheid. While most of the youth today are as Nkrumah (2021a) exclaims 'born frees' from social oppression, they are not exempt from the prevailing social conditions that exist because of that, i.e., inequality, lack of access, and lack of employment opportunities, nor the forthcoming and ongoing climate crisis (Nkrumah, 2021a).

Our results support the argument made by Marquardt et al. (2024), that young NSAs can contribute to the re-politicisation of climate change governance. This offers evidence that youth, as an oft-overlooked subgrouping of NSAs engaging in climate change governance, have unique capacities and ideas which other NSA groups could learn from. In doing so, it makes a valuable contribution to the political science literature on the role of NSAs in the Post-Paris era of climate governance in which adaptability and

experimentation are encouraged as key component of polycentric climate governance (Dorsch & Flachsland, 2017; Jordan et al., 2015).

This finding also demonstrates the value of intergenerational learning, thus contributing to the pedagogical literatures on policy-oriented and transformation-oriented learning in demonstrating that by disrupting the traditional model of one-way knowledge transmission from older to younger generations, we can build connections which contribute towards societal transformation whilst simultaneously developing skills to enhance policy implementation (Lotz-Sisitka, 2017; Macintyre et al., 2018).

### 4.2. Transformative/Transgressive Learning

Similar to Kowasch et al. (2021), who found participation in FFF protests enabled transformative learning by enhancing environmental citizenship, we found that an engagement in activism has enabled transformative learning in young South Africans while simultaneously providing renewed motivation for further engagement in collective action. In Figure 1, we have depicted this interconnection between extrinsic and intrinsic motivations, the overlap of intergenerational justice as both types, and motivations having a reciprocal and reinforcing relationship with transformative learning. We therefore support the argument that through a greater engagement with social movements, climate change education can have a transformative impact on both the learners and the societies they live in (Lotz-Sisitka et al., 2015; Macintyre et al., 2018).

It is important to be cognizant of Verlie and Flynn's (2022) line of questioning—if education is part of the system that youth seek to change, how can education itself contribute to transformation? We believe that one answer to this lies in developing climate change education which supports learners to develop the competencies to navigate and/or overcome barriers to engagement in collective climate action.

The most significant barriers which youth activists were able to overcome in this research were underpinned by either gaining access to knowledge—whether it be technical, scientific, or policy knowledge, or establishing their legitimacy and agency based on their lived and activist experiences. Climate change education should therefore support young people to increase their policy literacy while also striving to overturn intergenerational hierarchies. On this latter point, we support the calls to challenge colonial–capitalist pedagogies that inhibit transformation (Lotz-Sisitka et al., 2015; McGregor et al., 2018; Trott, 2024; Verlie & Flynn, 2022).

The motivations of youth participation clearly indicate that many youths learn about climate justice by engaging within policymaking and activist spaces rather than in formal education, and that many draw upon this experience to become educators themselves, mirroring Trott's (2024) findings in the USA. This leads us to view activism *as* education rather than as distinct from it and builds upon Wiek et al.'s (2012) argument that learning can and should occur with society rather than solely for society. Many youth participation frameworks and pedagogies explicitly or implicitly perpetuate the notion of a spectrum in which adults hold power and knowledge and young people can either be passive recipients or try to wrestle control from older people. By categorising barriers to youth participation, we find that youth are challenging this oversimplification, showing that in many instances, young people's engagement in dissent (O'Brien et al., 2018) interrupts this zero-sum framing of power. This lends weight towards the reconceptualization of education as a collaborative, dynamic, and reflexive way in which generations can work together to deliberate and co-produce participatory justice (Lotz-Sisitka et al., 2015).

While our findings align with Trott (2024) in demonstrating that youth activists can be educators, systemic barriers remain which prevent them from being recognised as such (McGregor & Christie, 2021). We find that where young South Africans are taking on

the mantle of education, they have become the teachers and are shifting perspectives on who can teach who, and who learns, creating a bi-directional flow of information (Trott, 2024; Verlie & Flynn, 2022). This aligns with McGregor and Christie (2021)'s finding that Scottish activists feel they can and should contribute to climate justice education, though the ambivalence of teachers in their study to accept this suggests a need to step beyond the confines of the classroom if we are to facilitate collaborative, transformative learning.

Our findings on the importance of context-specific awareness as a motivation for young climate activists supports the proposal from Kowasch et al. (2021) that climate change education should incorporate local, situated case studies into the curriculum to help learners connect to their own lived experiences. Recognising the importance of intrinsic motivations, we also emphasise Maria Ojala's important work (e.g., Ojala, 2012) in calling upon educators to create spaces for reflection on emotions to support students to constructively channel their emotions.

McGregor and Christie (2021) found activism and civic education could act as an education process and that climate justice education could learn from youth-led movements. Simultaneously, Kowasch et al. (2021) argued that activist spaces create alternative learning environments, and through teaching others, it inevitably enables transformative learning. Here, we suggest the use of and support for experiential learning experiences inside and outside the classroom as an avenue for youth to engage in climate activism and policymaking. Where Macintyre et al. (2018) highlight that policy literacy is a critical but often missing element in education, our findings suggest that youth-led activism is already helping to fill this gap.

## 5. Conclusions

This study explores youth motivations for and barriers to their climate activism within national and sub-national levels in South Africa. Guided by our lead-author's long-term engagement with South African youth climate networks and drawing upon in-depth semi-structured interviews with young activists who engage in a variety of climate governance spaces, we offer rich empirical contributions to start filling the research gap relating to youth activism in Africa and across the Global South. We then reflect upon the implications of these findings for climate change education and offer a series of provocations for policy-oriented and transformation-oriented pedagogies.

To explore why they participate, we explored their intrinsic and extrinsic motivations, finding that youth climate activists in South Africa are motivated by many of the same factors identified in other contexts, though, in contrast to their peers in the Global North (Cologna et al., 2021; Haugestad et al., 2021; Martiskainen et al., 2020), they particularly emphasise lived experience and what we refer to as "situated/context-specific awareness" (Figure 1). This provides an early indication that as members of the global youth climate movement increasingly have first-hand experiences of a climate-changed world, we could expect their activism to be strengthened and sustained, rather than undermined and eroded. This leads us to call for a greater inclusion of local, context-specific case studies in climate change education to motivate further engagement.

We also find that young South Africans use their policy and political participation to emphasise the intersectionality of climate, age, gender, and racial justice. Similarly to other NSA groupings (Keck & Sikkink, 1999; Carter, 2018), young people can therefore be seen as important contributors to climate change governance who engage in issue-framing and norm-setting in concerted efforts to re-politicise climate discourse (Marquardt et al., 2024). This leads us to call for an increased recognition of youth as holders of valuable knowledge and skills and for the provision of multi-directional, intergenerational learning.

Ultimately, we conclude that youth engagement in climate change policymaking and politics has significant transformative potential, and should be promoted and studied as key sites of learning, especially in contexts where young people are already experiencing climate change and living through it.

**Author Contributions:** Conceptualization, T.B. and H.T.; methodology, T.B.; software, T.B.; formal analysis, T.B.; investigation, T.B.; data curation, T.B.; writing—original draft preparation, T.B.; writing—review and editing, T.B. and H.T.; visualization, T.B.; supervision, H.T. All authors have read and agreed to the published version of the manuscript.

**Funding:** This research received no external funding.

**Institutional Review Board Statement:** The study was conducted in accordance with the University of Leeds Research Ethics Policy, and approved by UNIVERSITY OF LEEDS Research Ethics Committee through block ethical approval (reference AREA 20-070, April 2022).

**Informed Consent Statement:** Informed consent was obtained from all subjects involved in the study.

**Data Availability Statement:** The data are not publicly available as participants have not explicitly given permission for this data to be shared beyond the authors of this article.

**Acknowledgments:** This research formed part of the first author's MSc programme, made possible by Chevening Scholarships, the UK government's global scholarship programme, funded by the Foreign, Commonwealth and Development Office (FCDO) and partner organisations.

**Conflicts of Interest:** The authors declare no conflicts of interest.

## Appendix A

**Table A1.** Demographics of participants.

| Pseudonym | Gender | Age Group | Province |
|---|---|---|---|
| Nolwazi | Female | 18–24 | Gauteng |
| Themba | Male | 18–24 | Gauteng; Western Cape |
| Anaya | Female | 25–34 | Gauteng |
| Daniel | Non-binary | 18–24 | Western Cape |
| Amir | Male | 25–34 | Gauteng |
| Rania | Non-binary | 18–24 | Gauteng; Western Cape |
| Lerato | Female | 18–24 | Limpopo; Gauteng |
| Kgosi | Male | 18–24 | Mpumalanga |
| Mthunzi | Male | 25–34 | Kwa-Zulu Natal |
| Tumelo | Male | 18–24 | Gauteng |
| Ameena | Female | 18–24 | Western Cape |
| Clara | Female/non-binary | 25–34 | Western Cape |

## Appendix B

Semi-Structured Interview Guide

1. Can you tell me about your climate activism?
   Where and when did your activism start?
2. What (youth-led) projects have you been involved in?
   What was your role?
   How did you connect to/organise with other young people?

3. Are there any partnerships/networks between youth organisations for these projects?
4. What are the main themes/goals which you(th) are working on/advocating for?
   Who is it for?
   Why is it needed?
5. How have you been involved in any of governmental policy processes?
   At the local, national, and international level?
   Examples: Draft Climate Change Bill, NDC, Just Transition Framework, UNFCCC COP, Climate Action Plans
6. Do you think your concerns as a young person are reflected in current climate policies?
   Have you seen any tangible impacts from young people's lobbying/campaigns?
7. What are the challenges you have faced in participating?
   Do you think these challenges are unique to young people?
8. How would you define meaningful youth participation?
   How would that look?
   What would be the process?
   In your opinion, is the model near to that or does it need work?
9. Is there anything else you would like to talk about today? Anything you think I've missed in the questions above?
10. Do you have any questions for me?

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
