# Peer review of "Experiencing Climate Change and Living Through It—Provocations for Education Based on South African Youth Experiences of Climate Change Policymaking and Politics"

_2673-995X, doi:10.3390/youth5020037_

Round 1
Reviewer 1 Report
Comments and Suggestions for Authors
Thanks you for conducting this interesting an important research. The paper submitted could make a valuable contribution if revised to address key concerns.
1. Introduction: A good introduction to the research. Once the key arguments of the paper have been clarified, there are few minor considerations that should be made in the introduction. Firstly the first few lines chacterises climate change, biodiversity and pollution as "one" singular challenge. This reads strangely as I feel they are plural challenges. 2. Young people are conceptualised as "non-state actors". This term isn't used later in the paper. Perhaps the paper might clarify which "youth perspectives" it is exploring. The first paragraph, read in conjuction with the second, also suggests that governments have more limited technical and financial capacities that youth NSA. If so this needs further clarification.
2. Literature Review: This section draws on alot of interesting and relevant literature. A lot of different frameworks and concepts are introduced and the review itself is very broad. It might help either to narrow down the bredth of the paper or otherwise to tie together these different frameworks into one created by the authors.
3. Methdology: Further details could be provided here. It seems that belonging to a climate organisation was a key factor in determining participation. It is not in the participant profile table however. Further detail on recruitment and character of particpants would be helpful. More information is needed on the interviews. How long, what was the nature of the questions, when and where? More is needed on data analysis. How were the codes developed in Nvivo. The data analysis section feels a bit theoretical and more exact description of the process would be helpful.
Results: How were the results derived from the data. Again these cover alot and draw on different frameworks. It would be helpful to narrow the scope or pull together the sections into a more cohesive argument. There are a lot of quotations from the data but futher discussion of these would support the exploration. Further discussion of the comparison of intrinsic and extrinsic motivation is needed as the contrast between the two is not clear to me. Secion 3.3.1 feels dropped in. It either needs to be more of a focus and intergrated or removed. Figure 2 is very like Laura Ludy's model of child participation and would be worth looking at. Lundy, L. (2007). ‘Voice’is not enough: conceptualising Article 12 of the United Nations Convention on the Rights of the Child. British educational research journal, 33(6), 927-942.
Discussion: Again the results are discussed against multiple frameworks: different concepts of justice, intrinsic v extrinsic motivation, Anderson and O'Brien frameworks, litigation. Ensure the key frameworks are properly adressed in the lit review and then further develop. Some interesting findings are highlighted however.
Author Response
Thank you for taking the time to read our manuscript and for providing insightful and constructive feedback.
Please see the attached word document for details of how we have responded to each of your comments.

Reviewer 2 Report
Comments and Suggestions for Authors
Overall the piece is really interesting and adds a lot of perspectives on the area. It comes across however as a bit meandering and unwieldy, with too many strands and not an overarching point. it needs a lot of tightening.
Age range issue-
'This study follows the South African National Youth Policy (NYP) in defining youth as 14 to 34 years [42].'- this seems to be really stretching the dfn very old. It's of note that interviewees are all over 18.
WHile I can see the need to stick to over 14s or whatever, I dont understand this logic at all-
'While we acknowledge the importance of research into children’s participa- 311 tion in climate governance, we do not wish to further exacerbate the misrepresentation of 312 younger generations as homogenous. This study therefore only includes those who self- 313 define as youth rather than as children.' could there be a better explanation?
Typo -'One explanation for the disregard of youth in the climate governance literature is that 111 they have historically been perceived as subjects rather than agents'
It seems very long - does it meet word limit?
Author Response
Thank you for taking the time to read and comment on our manuscript.
Please see details of how we have responded to each of your comments in the attached word document.

Reviewer 3 Report
Comments and Suggestions for Authors
The study investigates the participation of South African youth in climate change politics and activism. It explains the motivations for youth climate activism through intrinsic and extrinsic factors such as lived experiences, context-specific awareness, and emotional responses. The study highlights how young activists frame and claim climate justice, emphasizing connections to social justice and intersectionality. It also analyzes how youth engage in climate policies through various projects and mobilizations. The research refines existing theoretical frameworks and proposes new categories, making significant contributions to the literature on youth participation.
The manuscript titled "Experiencing Climate Change and Living Through It - South African Youth Perspectives on Activism, Action and Justice" explores youth participation in climate change politics and policymaking in South Africa. The title and abstract are clear and descriptive, summarizing the research objectives, methodology, key findings, and conclusions. The introduction discusses the triple planetary crisis and the importance of including youth in climate governance, identifying a gap in Global South-facing studies. The literature review is comprehensive, covering climate change governance, youth participation, and climate justice, and combines two theoretical frameworks. The methodology employs qualitative research with semi-structured interviews of 12 young activists, using purposive sampling and detailed data collection and analysis processes. The results highlight intrinsic and extrinsic motivations for youth activism, perceptions of climate justice, and various modes of participation. The discussion interprets the findings, linking them to broader literature and emphasizing the unique contributions of South African youth. The manuscript offers theoretical contributions and practical insights for policymakers, suggesting areas for future research. Overall, the manuscript is well-structured with insightful findings, though it could benefit from a larger sample size and more detailed discussion on limitations and biases.

Author Response
Dear Reviewer 3,
Thank you for taking the time to read our manuscript, we are delighted to hear that you enjoyed it and thank you for your insightful and constructive feedback.
In response to your comment that: "Overall, the manuscript is well-structured with insightful findings, though it could benefit from a larger sample size"
We have discussed this in detail and do not feel that it would be appropriate to conduct more interviews at this time. We are comfortable that we reached saturation with the interviews we have and find that the data we collected from twelve participants is so rich and insightful that we are already struggling to keep the paper to an appropriate word limit. In addition, since these interviews were conducted, the context in South Africa has shifted from a single party national government to a multiparty coalition government making it difficult to ensure that insights from new interviews would be comparable.
In response to your comment that the manuscript could benefit from "a more detailed discussion on limitations and biases"
We have added to the limitations section of our methodology (lines 363-368). We feel that bias is sufficiently addressed in section 2.2 on positionality, however, we welcome your insights and if you think we have missed something we will happily address it.
Thank you once again,
the authors
Round 2
Reviewer 1 Report
Comments and Suggestions for Authors
Overall
This is much improved however- it still feels like there is a lot in it and I would suggest going deeper and limiting the scope. I think you could split this into two papers. Possibly the key areas in this paper are the motivations and challenges to participation and the implications for education. I think you could make the nature of the participation a separate article, unless you wanted to discuss different motivations as they relate to different types of participation. For example your results focus on motivation but these different types of motivation are not really discussed in your literature review- only giving passing reference.
The frameworks on the nature of participation- Andersson and O’Brien could then be pulled out for a separate article- with your findings and discussion on this explored in more detail.
Introduction
Great very clear. Couple of typos on page 2
Lin 36 scholars is repeated
Line 38 should it be “the” Global South.
It might help the reader to conclude the Introduction with a short sentence outlining what this paper does and you could also sign post the structure
Literature Review
If you can focus the article on motivations, challenges and the implications for education I would suggesting restructuring the lit review to include:
1.Defining youth (as you have)
- Understanding youth participation (all of which you have and it includes:
- Four research agendas in youth participation literature as is currently in the first paragraph under 1.2- you need to then flag which of these agendas you are covering.
- you could incorporate your literature on youth participating frameworks into a broader paragraph defining participation as including projects and actions with these different dimensions.
- links between participation and justice (as you have)
- You need to add more discussion of the literature on motivation and education into these sections if you are drawing on it in your analysis and results.
- And then gaps in existing research ( as you have and they include the paragraphs on
- Limited qual and the need for qual
- Importance of Global South perspectives
-Research in SA context (as you have)
-Your research question etc.
If you want to keep more to your existing paper and structure then you still need to tie together the existing section. So how do the frameworks, the existing research and the link to justice fit with the four research agendas outlined at the top of 1.2. If each of the subsequent sections cuts across the four agendas or if you are focussing on one or two- it would be helpful to be explicit with this.
- Methods and Materials- this is much clearer
Page 7 line 14 should it be while instead of which; line 46 is it “political processes or who were.
Results
I appreciate that you want to include lots of the participants' voices- however it would still be good to have more meta discussion on results.
If you keep Table 2 (I would suggest it is a separate paper)
- How was the categorization of the projects in table 2 done- was it straight forward, were there ambiguities. Can you explain these different approaches and why they are one or the other. What reflections do you have from this? As I say above I think this would be better as a separate paper.
Motivations- I would give more discussion between quotes. Let us know how typical the sentiment was using words like “many of the participants”, “one of the participants” etc. Did any of the data contradict each other. The learning through action is a key finding- but more needs to be in the lit review on this. Figure 1 is helpful- again more discussion would be helpful.
The discussion will be strengthened by addressing the literature in the lit review and then revisiting it here.
